

# Spot the bot: the inverse problems of NLP

Vasilii A. Gromov, Quynh Nhu Dang, Alexandra S. Kogan and Assel Yerbolova

HSE University, Moscow, Russia

## ABSTRACT

This article concerns the problem of distinguishing human-written and bot-generated texts. In contrast to the classical problem formulation, in which the focus falls on one type of bot only, we consider the problem of distinguishing texts written by any person from those generated by any bot; this involves analysing the large-scale, coarse-grained structure of the language semantic space. To construct the training and test datasets, we propose to separate not the texts of bots, but bots themselves, so the test sample contains the texts of those bots (and people) that were not in the training sample. We aim to find efficient and versatile features, rather than a complex classification model architecture that only deals with a particular type of bots. In the study we derive features for human-written and bot generated texts, using clustering (Wishart and K-Means, as well as fuzzy variations) and nonlinear dynamic techniques (entropy-complexity measures). We then deliberately use the simplest of classifiers (support vector machine, decision tree, random forest) and the derived characteristics to identify whether the text is human-written or not. The large-scale simulation shows good classification results (a classification quality of over 96%), although varying for languages of different language families.

# INTRODUCTION

The rapid evolution of artificial intelligence (AI) drives humanity to the post-truth world (We may see its emergence now). Generative AI (text bots, large language models, *etc.*), an important driver of the process, tends to distort human linguistic personality (language aspect of human identity). Currently, the linguistic personality evolves, influenced by texts written by humans (mainly, with good intentions)—from childhood fairy tales to university calculus textbooks. With recent advances in text generation, we will soon find ourselves immersed, from birth, in a gigantic ocean of texts ('with a natural-identical flavour'), generated by bots (Evidently, bots can generate texts at a much greater speed than humans can write them) distorting a child's linguistic personality. This makes it necessary to design AI systems able to distinguish bot-generated and human-written texts (Spot the Bot).

Frequently, detection methods try to 'spot' a particular bot, which does not seem highly reasonable. This article is a continuation of the previous work *Gromov & Dang (2023a)* and *Gromov & Kogan (2023)*, in which structural differences of texts, generated by different bots and human-written texts were analysed. In the present article, we expand the study

Corresponding author
Alexandra S. Kogan, akogan@hse.ru

and examine other languages, as well as other text bot models. Most importantly, we focus on the task of finding efficient features, that could be used to identify texts, generated by different types of bots (rather than constructing a classification model that can only detect a specific bot). For this reason, we propose a modified problem statement: for a given natural language, separate texts written by people from all texts generated by bots. To check the performance of the respective classifier and its ability to generalise, we propose to randomly divide the set of bots into bots used to construct the classifier and those not used; the latter is employed to generate texts for a test set. Consequently, in order to construct the classifier, one should utilise the most general features of the semantic space. In addition, one should validate the respective hypotheses for several languages, preferably those of various language groups and families.

In the study, we derive features for human-written and bot generated texts, using clustering (Wishart and K-Means, as well as fuzzy variations) and nonlinear dynamic techniques (entropy-complexity measures). We then deliberately use the simplest of classifiers (support vector machine, decision tree, random forest) are the used to distinguish human-written texts from bot-generated texts. In order to construct classifiers, we consider the following hypotheses:

1. For the space of word embeddings, the areas 'haunted' by bots and those 'visited' by humans coincide—if anything, people and bots share the same dictionary. On the contrary, for the space of $n$-grams, one can reveal human and bot areas: humans tend to produce unexpected sequences of words more than bots (people write more sophisticatedly, 'spicier').

2. Therefore, a large number of $n$-grams corresponding to bot areas, and a small number of $n$-grams corresponding to human ones betrays a bot text.

3. Crisply clustered, bot-generated texts yield more compact clusters of $n$-gram embeddings, as compared with those for human-written texts. Also, the characteristics of human and bot clusterings are statistically significantly different from each other.

4. Fuzzily clustered, bot-generated texts yield less fuzzy clusters, as compared with those for human-written texts.

5. The non-linear dynamics characteristics of the semantic trajectories statistically significantly differ for human-written and bot-generated texts.

Natural sciences consider two kinds of problems—the direct problems and the inverse problems (*Groetsch & Groetsch, 1993*). The direct problem means that one seeks to solve a clearly formulated problem. On the contrary, the inverse problem means that one reconstructs the problem from observations, determining its characteristics. By analogy, we propose to divide natural language processing problems into the direct and inverse ones, with pairs like 'to design a bot—to detect a bot,' 'to translate text—to automatically assess translation quality,' *etc.*

The rest of this article is organised as follows. The next section reviews recent advances in bot detection. The third discusses proposed methods of bot detection; the fourth analyses the classifiers performance. We discuss conclusions and further directions in the final sections.

# RELATED WORKS

Many articles on the bot detection problem centre on metadata analysis (information on bot accounts, their profiles, the dynamics of message generation, and other features not associated with the texts themselves). It is also possible to use information about interactions between accounts, which can be represented as a graph. *Daya et al. (2019)* proposed a graph-based bot detection system for communication graph. *Mesnards et al. (2021)* found out that social media bots tend to interact more with humans than with other bots and this feature (heterophily) can be used to detect them. BotFinder *Li et al. (2022)* uses Node2Vec *Grover & Leskovec (2016)* embeddings and community detection to identify bots in social media. *Pham et al. (2021)* proposed Bot2Vec –an improved Node2Vec *Grover & Leskovec (2016)* algorithm for the bot detection task. There are various models which utilize graph convolutional networks (GCN): BotRGCN *Feng et al. (2021)*, SqueezeGCN *Fu et al. (2023)*, SEGCN *Liu et al. (2024)*. *Latah (2020)*, in his review, provides a detailed classification of such methods. The author concludes, based upon the Defense Advanced Research Projects Agency (DARPA) bot identification competition, that 'bot detection had to be semi-supervised', that is, algorithms of this class should be verified and corrected by humans.

In this study, in order to solve the problem of bot detection we employ only texts generated by bots and written by humans. Usually, the research articles on the algorithms of this class tend to study a single (a group of) specific bot in order to build a kind of anti-bot, usually using a neural network model. For example, *Garcia-Silva, Berrio & Gomez-Perez (2021)*, to identify Twitter bots, employ fine-tuned generative pre-trained transformers (GPT, GPT-2); pretrained embeddings: global vectors for word representation (GloVe), Word2Vec, fastText, and contextual embeddings from language model (ELMo) are also used (*Garcia-Silva, Berrio & Gomez-Pérez, 2019*).

We believe that the statement of the bot detection problem considered in this article—to draw the demarcation line between all texts written by people and all texts generated by bots—is the most reasonable one, both practically and theoretically. Unfortunately, little or no work deals with the problem of bot detection in this particular statement. At the same time, one can refer to a number of articles that, without directly touching on the problem, develop methods to explore the structure of the semantic space of a natural language, furnishing one with theoretical grounds to solve the problem in question.

Here, the first approach concerns itself with the simplest characteristics of the corresponding texts: for example, *Kang, Kim & Woo (2012)* employ simple lexical and syntactic features, such as the frequency of letters or the average length of a word. *Cardaioli et al. (2021)* model a Twitter user using a set of stylistic features, and distinguish bot and human accounts by estimated consistency of their message style. *Chakraborty, Das & Mamidi (2022)* combine text feature extraction with graph approaches. *Gromov & Dang (2023b)* transform text into a multivariate time series to calculate its entropy and complexity in order to distinguish bots and humans' time series. *Hernandez-Fernáandez et al. (2022)*, (using the example of the Catalan language) summarise the basic laws inherent in texts of natural languages at the morpheme-word level. Importantly, the overwhelming majority

of the laws are power ones; *Torre et al. (2019)* provide the psychophysiological reasons for these laws to appear (see also *Baixeries, Elvevag & Ferrer-i Cancho (2013)*, *Gromov & Migrina (2017)*). *Wang & Liu (2023)* indicate that a power law exponent in Zipf's law constitutes a measure of lexical diversity.

The second approach utilises sentiment analysis of texts in order to distinguish human-written and bot-generated ones: their sequences of emotional characteristics tend to differ from each other. This approach seems to currently exhibit a surge in publications (*Uymaz & Metin, 2022*; *Monica & Nagarathna, 2020*). *Heidari, James Jr & Uzuner (2021)* employ complex sentiment features of tweets (in English and Dutch). *Liao et al. (2021)* propose a novel multi-level graph neural network (MLGNN) for text sentiment analysis; *Lin, Kung & Leu (2022)* and *Galgoczy et al. (2022)* combine Bidirectional Encoder representations from transformers (BERT) with a text sentiment analysis to identify harmful news. Also promising are predictive clustering methods (*Lira, Xavier & Digiampietri, 2021*; *Gromov & Baranov, 2023*; *Gromov & Borisenko, 2015*; *Gromov & Konev, 2017*; *Gromov & Shulga, 2012*), which reveal characteristic subsequences (motifs) in a time series. Many articles use labelled data to train neural networks: for instance, *Mu & Aletras (2020)* develop a dataset of Twitter users with retweets from unreliable or reliable news sources, *Ren & Ji (2017)* discuss an efficient model to detect spam with a false opinion. Their model demonstrates good results; however, the authors point out the limitations of supervised learning methods and the need for research into unsupervised ones.

Tanaka-Ishii, in her monograph (*Tanaka-Ishii, 2021*) and a series of articles (*Tanaka-Ishii & Aihara, 2015*; *Tanaka-Ishii & Bunde, 2016*; *Tanaka-Ishii & Takahashi, 2021*), examines long correlations between words in the text (see also (*Altmann & Gerlach, 2016*; *Altmann, Cristadoro & Esposti, 2012*)), seemingly, a promising feature to distinguish human and bot texts. Apparently, it would be extremely difficult for bots (even trained using very complex neural network models) to feign this kind of sequence (since the majority of them learn from local information and learn to track only local connections within one $n$-gram).

*Brown et al. (1992)* examine the entropy characteristics of a natural language. *Debowski (2020)*, in his monograph, explores the main characteristics of natural languages; he establishes that most characteristics follow power laws. *Gromov & Konev (2017)* examine a natural language as a whole (For the Russian and English languages); they reveal that lengths of texts written in a particular natural language satisfy a power-law distribution; here, one can detect a bot using not a single text, but many texts simultaneously. To sum up, these language laws offer promise as bot-detecting tools—it would be difficult for a bot to generate texts complying with all the laws discussed in these works. The above approaches deal with sequences of characters in a language. It seems to be more reasonable to deal with sequences of meanings, given by their embeddings, in the semantic space—semantic trajectories (*Gromov & Dang, 2023b*). It is possible to analyse semantic trajectories as those of dynamic systems (*Malinetskii & Potapov, 2000*; *Kantz & Schreiber, 2004*) and to establish information-theoretic characteristics of these trajectories (*Gromov & Dang, 2023b*).

To summarise, in natural languages, one can broadly distinguish two groups of features: local and global (holistic). Local features rely on properties of individual words or $n$-grams;

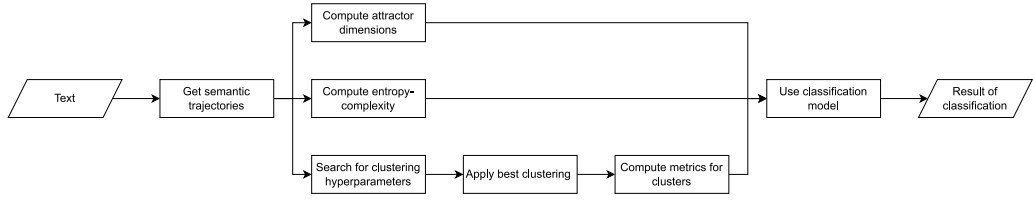

**Figure 1** Methodology flowchart.

global properties, on those of an entire text or even a language itself. Frequently, local characteristics fail to distinguish human-written texts and bot-generated texts, whereas global ones can do the job (Comte de Buffon's maxim 'Style is the man' seems to acquire a new meaning here).

## PROBLEM STATEMENT

For a given natural language, one considers a space of texts $\Omega$, both written by humans and generated by bots. The space is divided into a subspace $A = \{\alpha_1, \ldots, \alpha_a\}$ of the texts written by humans and a subspace $M = \bigcup_{j=1}^{L} M_j$ of texts generated by bots (Cf. Ancient Greek ἄνθρωπος (a man) and μηχάνημα (a machine).). $M_j = \{\mu_1, \ldots, \mu_{m_j}\}$ comprises text generated by the $j$-th bot. The objective is to construct a set features $\Lambda = \{\lambda_1, \ldots, \lambda_k\}$ and to build a classifier $R = R(\Lambda)$ with a classification F1-score above threshold $r*$.

One randomly samples human-written texts in order to construct training and test sets. Most importantly, in order to construct training and test sets for bot-generated texts, one does not randomly sample this set of texts, but randomly samples the set of bots themselves $\{M_j, j = 1 .. l\}$ into training and test subsets. The former comprises bots generating texts used to train the classifier; the latter, bots generating texts used to test it. The number of texts and the distribution of text sizes are approximately the same as those for the training and test sets for human-written texts.

## METHODS

A flowchart of the text classification process is shown in Fig. 1.

### Data collection and preprocessing

In order to construct and verify universal bot detection models, we conduct simulations with the employment of texts of various languages for various language families, as summarised in Table 1. For each language, Table 1 provides information about its language group and family; the number of human-written texts in the sample; and the average text size. To collect human-written texts, we use the national literature corpora: we believe it is the national literature that embodies the language and associated language processes best. All texts are collected from open sources (Project Gutenberg and so on). For analysis, we employ only texts of 100 words and more. Each text is tokenised and lemmatised (Table 1 summarises the lemmatisation models used). In addition, we replace pronouns,

**Table 1  Human-written corpora details for Russian, English, German, Vietnamese and French languages.**

| Language | Group | Family | # texts | Avg. text length |
|----------|-------|--------|---------|------------------|
| Russian | East Slavic | Indo-European | 6,429 | 14,510 |
| English | Germanic | Indo-European | 11,052 | 21,744 |
| German | Germanic | Indo-European | 12,503 | 72,878 |
| Vietnamese | Vietic | Austroasiatic | 1,071 | 54,496 |
| French | Romance | Indo-European | 8,405 | 66,946 |

prepositions, numerals, and proper nouns with the respective tokens, using named entity recognition and part of speech tagging models.

We employ four bot types of varying complexity to create an effective algorithm for simple and complex models: long short-term memory (LSTM), GPT-2, multilingual GPT (mGPT), "Yet Another Language Model" (YaLM). We provide some details on the models hyperparameters, sizes and text generation process in Appendix A.

## WORD EMBEDDINGS

To obtain embeddings, we use two approaches: the first employs singular value decomposition (SVD) of the term frequency-inverse document frequency (TF-IDF) matrix for words and texts (*Bellegarda, 2022*); the second, Word2Vec (*Mikolov et al., 2013*), a neural-network model. The methods are widely used to study the semantics of texts: both SVD and Word2Vec embeddings capture structural relationships between words. We provide some details on the word embedding techniques in Appendix B.

We obtain embeddings for $n$-grams by concatenating the ones for words of this $n$-gram.

Thus, to construct a sample of $n$-gram embeddings, one should (1) collect a corpus of natural language texts; (2) pre-process them; (3) build a dictionary (a set of words with corresponding embeddings); (4) build an $n$-gram dictionary for all $n$-grams of the language.

### *Semantic trajectories*

Several approaches used in this article rely on the concept of the semantic trajectory of a text. By a semantic trajectory we mean a sequence of word embeddings (*Gromov & Dang, 2023b*), thought of as a multidimensional time series. Gromov and Dang reveal the chaotic nature of semantic trajectories for texts of literary masterpieces for the Russian and English languages.

### Characteristics for text bot detection

This section discusses the characteristics the classifier employs for bot identification. Since the classifier itself is extremely simple, the performance of the bot-detection algorithms depends mainly on the characteristics they use.

### *Entropy and complexity of semantic trajectories*

Martin, Plastino, and Rosso (*Rosso et al., 2007*) proposed a method to distinguish chaotic series from simple deterministic processes, on the one hand, and stochastic processes, on

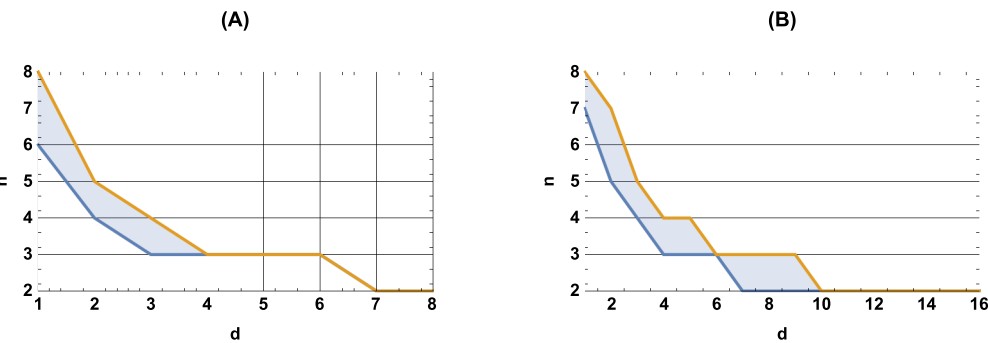

**Figure 2 Admissible values of n and d for the Russian (left) and English (right) languages.** For n and d values above the orange boundary texts fall into the deterministic area; below the blue boundary - into the noise. From *Gromov & Dang (2023b)*.

the other. The method employs entropy and complexity of a time series. The procedure for calculating entropy and complexity is presented in Appendix C.

This method allows one not only to build a classifier, but also (which is important for all methods discussed in the article) to establish the values of *n* and *d* (the number of words in an n-gram, and the dimension of the embedding space, respectively) for which the semantic trajectory reflects the true 'dynamics' of the text. If the values of these quantities are too small, the semantic trajectory is mapped to a point on the entropy-complexity plane, which belongs to the area of purely random processes; as the values of these quantities increase, the point shifts to the region of chaotic processes; with a further increase, the point moves to the region of simple deterministic processes. We believe that the true dynamics of a text are reflected by those values of *n* and *d* at which the semantic trajectory is mapped into the area of chaos (refer to *Gromov & Dang (2023b)* for details). By way of illustration, Fig. 2 shows, for the Russian and English languages, the values of *n* and *d*, such that most semantic trajectories of the national literature corpus fall into the area of chaotic processes.

The hypothesis to be tested within this framework is as follows: 'For certain values of the number of words in an *n*-gram, *n*, and the dimension of the embedding space, *d*, the points on the entropy-complexity plane corresponding to the semantic trajectories statistically significantly differ for human-written and bot-generated texts.' Naturally, all pairs *n* and *d*, such that the hypothesis holds true, belong to the chaos region.

### Attractor dimensions of semantic trajectories

This approach also employs the semantic trajectories of texts: to solve the problem in question, one estimates various characteristics of a dynamic system (and its attractor) that generate observed (multidimensional) time series (semantic trajectories).

The hypothesis to be tested within this framework is as follows: 'For certain values of the number of words in an *n*-gram, *n*, and the dimension of the embedding space, *d*, the estimated dimensions of the dynamic systems attractors statistically significantly differ for human-written and bot-generated texts.' In this article, to this end, we use the Renyi

entropy of a dynamic system (*Malinetskii & Potapov, 2000*; *Kantz & Schreiber, 2004*). The procedure of calculating the attractor dimensions is presented in Appendix D.

### Clustering of n-grams and cluster cohesion measures

This section discusses features based on the large-scale, coarse-grained structure of the semantic embedding space. Interestingly, one cannot trace these differences (discussed below) for words ($n = 1$)—after all, bots use the same dictionaries as people do—but can trace this for bigrams, trigrams, *etc.* ($n > 1$)—people tend to produce more unexpected, non-trivial sequences of words. The hypotheses to be tested within this framework are as follows:

1. For crisp clustering, $n$-grams of bot-generated texts statistically significantly yield more compact clusters than those of human-written texts.
2. For fuzzy clustering, $n$-grams of bot-generated texts statistically significantly yield more clearly defined cluster cores and smaller fuzzy areas than those of human-written texts.
3. A large-scale simulation reveals areas of semantic space more frequently 'visited' by people, and those more frequently 'haunted' by bots. Respectively, the hypothesis to be tested within this framework is as follows: 'For bot-generated texts, $n$-grams belonging to the 'bot areas' appear statistically significantly more frequently as compared to human-written texts.'

All above hypotheses involve clustering $n$-gram embeddings, using one or another clustering algorithm. The clusters of $n$-gram embeddings tend to exhibit rather whimsical shapes; moreover, the total number of clusters is unknown *a priori*. This imposes demanding requirements on the clustering algorithm used. On the one hand, it should not require *a priori* knowledge of the number of clusters; on the other hand, it should allow clusters of various shapes. Whereas, in order to fulfil the first requirement, one may run the algorithm for various preset numbers of clusters (within a reasonable range); in order to fulfil the second requirement, one should employ specific clustering algorithms—it should be clearly understood that any clustering algorithm implicitly defines what it 'considers' to be a cluster.

In the present, we employ several clustering algorithms: K-Means (*MacQueen, 1967*); C-Means (*Bezdek, Ehrlich & Full, 1984*) (the latter is a fuzzy counterpart of the first); the Wishart clustering algorithm (*Wishart, 1969*); the Wishart algorithm with fuzzy numbers (*Novak, Perfilieva & Mockor, 2012*). The Wishart algorithm combines graph- and density-based clustering concepts: this combination makes it possible to identify clusters of almost any structure and determine their number in the course of the clustering process. We provide their pseudocodes and detailed descriptions in Appendix E.

With crisp clustering algorithms, we use the following characteristics to estimate the compactness of crisp clusters:

1. The number of elements (without repetitions) in a cluster: an $n$-gram can appear in texts, and thereby in a cluster, many times—however we count it just one time. The characteristic is normalised to the size of the largest cluster of the clustering.
2. The number of elements (without repetitions) in a cluster. The characteristic is normalised to the total number of unique vectors in the sample.

3. The number of elements (with repetitions) in a cluster: here, we count how many times the $n$-gram appears in the texts. The characteristic is normalised to the size of the largest cluster of the clustering (the size here is the number of elements with repetitions).

4. The number of elements (with repetitions) in the cluster. The characteristic is normalised to the total number of elements (with repetitions) in the clustering.

5. The maximum distance from a cluster element to the centre of the cluster.

6. The average distance from a cluster element to the cluster centre.

7. The maximum distance between elements in a cluster.

8. The average distance between elements in a cluster.

In all cases, the Euclidean distance is used. Description of fuzzy numbers and application of Euclidean distance to them are presented in Appendix F.

The above approaches imply various ways to cluster data, and thereby various clusterings, produced by both various values of hyperparameters for the same algorithm and different clustering algorithms. In order to compare the quality of the clusterings, we employ clustering quality measures (*Xiong & Li, 2018*).

## Training and test data

According to the problem statement, the set of bots $M$ is randomly divided into a set of bots used to train classifier models and that used to test it. A random selection gave: GPT2 and YaLM for training bots $(M_1, M_2)$, LSTM and mGPT for test ones $(M_3, M_4)$.

The training and test dataset sizes for all languages are identical: the training dataset consists of 2,000 human-written texts and 2,000 bot-generated texts, the test dataset consists of 600 human-written texts and 600 bot-generated texts. The distribution of text lengths for bots and people is also similar; for more details, see Appendix G.

We deliberately consider the simplest models for classification: support vector machine, decision tree, and random forest. The hyperparameters of each model are selected by 10-fold cross-validation. We set the F1-score threshold $r^* = 0.9$.

## CLASSIFICATION RESULTS OF NATURAL LANGUAGE TEXTS

The values for the embedding space dimension d and the number of words in the $n$-gram $n$ is of decisive importance for the efficiency of the algorithms under consideration. As noted earlier, as a criterion to choose the values of these two parameters, we consider that the points with entropy and complexity coordinates for most semantic trajectories of a corpus of literary texts fall into the 'region of chaos' on the entropy-complexity plane. A large-scale simulation, for all languages under consideration, established parameters for which the vast majority of literary texts fall into the region of chaos.

As an illustration, Fig. 3 exhibits points on the entropy-complexity plane that correspond to the semantic trajectories of Russian literary texts: it is clear that in the left subplot of Fig. 3 ($n = 4, d = 1$) the vast majority of points fall into the area of simple random processes (too small values); in the central subplot of Fig. 3 ($n = 3, d = 4$), into the area of chaotic processes (optimal values), in the right subplot of Fig. 3 ($n = 5, d = 4$), into the area of simple deterministic processes (too large values); refer to Appendix H for other languages

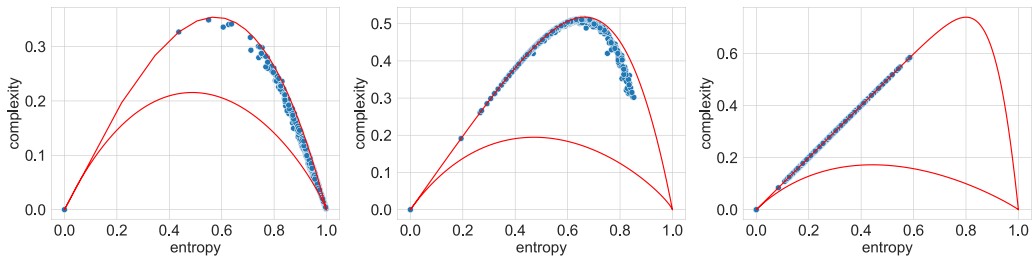

**Figure 3** **Entropy-complexity plane: points corresponding to texts of Russian literature.** For $n = 4$, $d = 1$ (left), the texts fall into the region of simple random processes; for $n = 3$, $d = 4$ (centre), into the region of chaotic processes; for $n = 5$, $d = 4$ (right), into the region of simple deterministic processes.

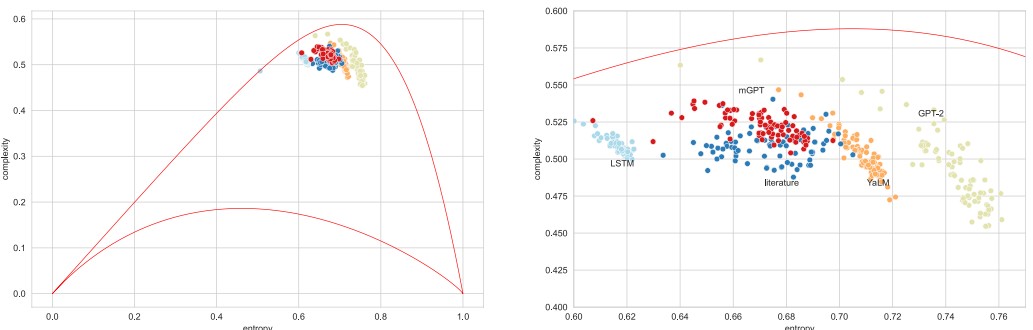

**Figure 4** **Entropy-complexity plane: points corresponding to texts of Vietnamese language for admissible values $n = 4$, $d = 3$ (full plane on the left, magnified part on the right).** Blue points correspond to literary texts, red—to texts generated by mGPT, beige—by GPT-2, light blue—by LSTM model, orange—by YaLM.

data. We think that smaller and larger $n$ and $d$, for which most texts seem to belong to the region of purely random and simple deterministic processes, respectively, do not reflect the nature of the texts (For larger n, we attribute this to the fact that we cannot adequately estimate entropy and complexity for very large values of $n$ and $d$, since trajectories are limited in size.).

It is of interest that the areas of optimal values can differ significantly for different languages; for example, for Vietnamese, the area under consideration includes longer sequences of words; for $d = 1$, the optimal values of $n$ are from 10 to 14; whereas for Russian—from 6 to 8, and for English—from 7 to 8. We attribute this with the word order in the language (free, fixed, intermediate options).

## Classification with entropy-complexity measure

Figure 4 presents typical sets of points on the entropy-complexity plane, corresponding to the semantic trajectories of literary texts. The picture corresponds to the Vietnamese language: here the blue dots correspond to literary texts, the dots of other colours correspond to texts generated by the bots in question—the picture allows us to conclude that it is possible to separate texts using these features.

**Table 2 F1-score values for classification with entropy-complexity measures.** SVM stands for Support Vector Machine; DT, for Decision Tree; RF, for Random Forest

|  | Russian | English | German | French | Vietnamese |
|---|---|---|---|---|---|
| SVM | 0.89 | 0.64 | 0.97 | 0.95 | 0.95 |
| DT | 0.76 | 0.81 | 0.97 | 0.59 | 0.96 |
| RF | 0.78 | 0.82 | 0.98 | 0.87 | 0.97 |

Table 2 summarises the results of classifiers based on these characteristics, for a test sample (for accuracy scores of the classifiers, see Appendix I). It is curious that the optimal models differ for different languages: for Russian and French, the highest classification quality is achieved with the support vector machine; for English, German and Vietnamese, with random forest. At the same time, the F1-score values of the optimal models are above 0.8, which indicates that even when trained on the texts of one set of bots (GPT2 and YaLM), the model has a generalisation ability and separates literary texts from texts generated by LSTM and mGPT, which it has not encountered before, when training a classifier.

## Classification with semantic trajectory characteristics

Figure 5 exhibits a characteristic view of the set of values of generalised entropy ($q = 0..20$) for the Vietnamese language. A reader may find similar figures for other languages and other values of $q$ in Appendix J. Table 3 presents the results of simulation for this type of feature(for accuracy scores of the classifiers, see Appendix I). The results obtained allow us to conclude that features of this type give significantly worse results than other types of features. However, even the worst results for German and Vietnamese languages here do not drop below 0.5. For the Russian, English, and French languages, the best appear to be decision trees trained on SVD embeddings; for the German language, a decision tree trained on CBOW vectors. Importantly, as in the case of entropy-complexity, the decision tree cannot be called a universally optimal model—for the Vietnamese language the decision tree is retrained, and the classification quality is higher when using the support vector machine (on CBOW embeddings).

## Classification with characteristics derived from the clustering of n-grams

Table 4 summarises the simulation results for features based on the Wishart clustering technique (refer to Sect. Clustering of $n$-grams and cluster cohesion measures, for accuracy scores of the classifiers, see Appendix I). For most languages, SVM shows significantly better results than a decision tree, but the decision tree works much better for the Russian language. The best embedding also depends on the language. Table 5 contains the corresponding results for features based on K-Means clustering (for accuracy scores of the classifiers, see Appendix I). Similarly, SVM produces better results than a decision tree for the majority of languages. However, for Russian and Vietnamese languages, the decision tree shows notably better F1 scores. The best embedding for K-Means clustering also cannot be chosen. Thus, it is impossible to select a specific embedding or a specific architecture for classification

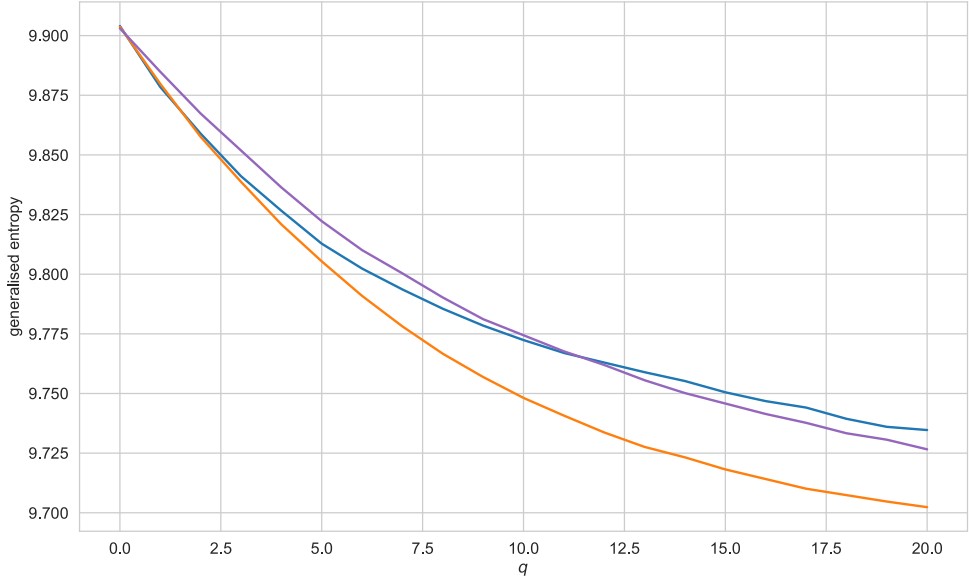

**Figure 5** **Generalised entropy values for texts of the Vietnamese language, *q* ranging from 0 to 20.** Blue line refers to literary texts, purple line - texts generated by GPT-2, orange - texts, generated by LSTM.

**Table 3** **F1-score values for classification with semantic trajectory characteristics.**

|  | Russian | English | German | French | Vietnamese |
|---|---|---|---|---|---|
|  | | | Support Vector Machine | | |
| SVD | 0.52 | 0.50 | 0.59 | 0.75 | 0.52 |
| CBOW | 0.00 | 0.59 | **0.77** | 0.00 | **0.70** |
| Skip-Gram | 0.00 | 0.00 | 0.65 | 0.59 | 0.50 |
|  | | | Decision Tree | | |
| SVD | 0.66 | 0.78 | 0.65 | 0.85 | 0.67 |
| CBOW | 0.02 | 0.58 | 0.80 | 0.00 | 0.62 |
| Skip-Gram | 0.00 | 0.00 | 0.62 | 0.66 | 0.59 |
|  | | | Random Forest | | |
| SVD | 0.68 | 0.79 | 0.63 | 0.86 | 0.68 |
| CBOW | 0.05 | 0.56 | 0.81 | 0.00 | 0.68 |
| Skip-Gram | 0.00 | 0.00 | 0.68 | 0.74 | 0.60 |

using clustering of *n*-gram text vectors. It is curious that for different languages, the best results were shown by different combinations of embedding type and classifier variant, and, most importantly, the variant that is the best for a given language shows the poor results for languages of other language families.

A large-scale simulation reveals that the classifier performance depends on the clustering algorithm used. Table 6 summarises classification results for clustering algorithms used:Wishart, K-Means and their fuzzy variations—Fuzzy Wishart and C-Means (for accuracy scores of the classifiers, see Appendix I). Intra-cluster distance features were used for classification (*i.e.,* features 5-8 listed in Sect. Clustering of *n*-grams and cluster cohesion

**Table 4  F1-score values for Wishart clustering-based classifiers.**

|           | Russian | English | German | French | Vietnamese |
|-----------|---------|---------|--------|--------|------------|
| Support Vector Machine | | | | | |
| SVD       | 0.41    | 0.75    | 0.92   | 0.93   | 0.57       |
| CBOW      | 0.53    | 0.87    | 0.60   | 0.57   | 0.76       |
| Skip-Gram | 0.61    | 0.81    | 0.58   | 0.96   | 0.82       |
| Decision Tree | | | | | |
| SVD       | 0.61    | 0.70    | 0.81   | 0.84   | 0.76       |
| CBOW      | 0.81    | 0.75    | 0.69   | 0.57   | 0.32       |
| Skip-Gram | 0.79    | 0.68    | 0.58   | 0.64   | 0.28       |
| Random Forest | | | | | |
| SVD       | 0.48    | 0.77    | 0.89   | 0.48   | 0.45       |
| CBOW      | 0.81    | 0.83    | 0.64   | 0.58   | 0.28       |
| Skip-Gram | 0.78    | 0.76    | 0.52   | 0.66   | 0.31       |

**Table 5  F1-score values for K-Means clustering-based classifiers.**

|           | Russian | English | German | French | Vietnamese |
|-----------|---------|---------|--------|--------|------------|
| Support Vector Machine | | | | | |
| SVD       | 0.59    | 0.78    | 0.92   | 0.58   | 0.70       |
| CBOW      | 0.21    | 0.95    | 0.82   | 0.84   | 0.44       |
| Skip-Gram | 0.40    | 0.86    | 0.85   | 0.85   | 0.26       |
| Decision Tree | | | | | |
| SVD       | 0.33    | 0.80    | 0.87   | 0.35   | 0.72       |
| CBOW      | 0.74    | 0.87    | 0.47   | 0.45   | 0.27       |
| Skip-Gram | 0.84    | 0.82    | 0.32   | 0.71   | 0.19       |
| Random Forest | | | | | |
| SVD       | 0.50    | 0.92    | 0.86   | 0.83   | 0.66       |
| CBOW      | 0.18    | 0.90    | 0.38   | 0.93   | 0.17       |
| Skip-Gram | 0.78    | 0.86    | 0.62   | 0.63   | 0.22       |

measures). In general, the Wishart algorithm (in its crisp and fuzzy versions) proved to be the best algorithm for identifying features for the classifier.

The introduction of fuzziness into the Wishart algorithm significantly improves the classification algorithm performance: it achieves the best results for German, French, and Vietnamese on random forest models trained on features extracted with the employment of its fuzzy modification. For both Russian and English languages, one can also observe the better performance of the model, as compared to the crisp Wishart algorithm. This is most clearly expressed for the French language—the crisp Wishart gave the value F1 = 0.61; whereas its fuzzy version, F1 = 0.93.

## Unified classification model

We also examine a classifier based on all the features discussed above. Table 7 summarises the classification results (for accuracy scores of the classifiers, see Appendix I). A simulation shows that mechanical combination of features does not necessarily improve classification

**Table 6** F1-score values for classifiers based on intra-cluster distances.

|  | Russian | English | German | French | Vietnamese |
|---|---|---|---|---|---|
| Support Vector Machine | | | | | |
| Wishart | 0.53 | 0.64 | 0.54 | 0.35 | 0.68 |
| Fuzzy Wishart | 0.49 | 0.84 | 0.50 | 0.89 | 0.60 |
| K-Means | 0.95 | 0.80 | 0.51 | 0.63 | 0.65 |
| C-Means | 0.93 | 0.76 | 0.52 | 0.47 | 0.54 |
| Decision Tree | | | | | |
| Wishart | 0.53 | 0.64 | 0.54 | 0.35 | 0.68 |
| Fuzzy Wishart | 0.49 | 0.84 | 0.50 | 0.89 | 0.60 |
| K-Means | 0.95 | 0.80 | 0.51 | 0.63 | 0.65 |
| C-Means | 0.93 | 0.76 | 0.52 | 0.47 | 0.54 |
| Random Forest | | | | | |
| Wishart | 0.55 | 0.72 | 0.71 | 0.61 | 0.67 |
| Fuzzy Wishart | 0.69 | 0.85 | 0.89 | 0.93 | 0.81 |
| K-Means | 0.98 | 0.86 | 0.61 | 0.51 | 0.70 |
| C-Means | 0.95 | 0.78 | 0.60 | 0.67 | 0.72 |

**Table 7** F1-score values for unified classification model. SVM stands for Support Vector Machine; DT, for Decision Tree; RF, for Random Forest.

|  | Russian | English | German | French | Vietnamese |
|---|---|---|---|---|---|
| SVM | 0.82 | 0.98 | 0.63 | 0.82 | 0.74 |
| DT | 0.76 | 0.85 | 0.55 | 0.59 | 0.62 |
| RF | 0.86 | 0.86 | 0.57 | 0.68 | 0.66 |

quality. For example, for German and Vietnamese, the optimal F1-measure values for the unified classifier are below 0.75, compared to the entropy-complexity-based classifier (F1 = 0.98 for German and 0.97 for Vietnamese). On the other hand, the quality of classification of English texts has significantly improved: the F1 value amounts to 0.98, as compared to the maximum 0.87 of the classifier based on the characteristics of the clusters. At the same time, it was found that the addition of some features reduces the differentiating ability of other features, and therefore the model as a whole.

## CONCLUSIONS

The present article states the problem of distinguishing all human-written and bot-generated texts—in our opinion, this formulation of the problem is more reasonable than the task of identifying an individual bot, no matter how effective and efficient it is. The training and test samples are formed by randomly separating not the texts of bots and people, but the bots and people themselves, so the test sample contains the texts of those bots (and people) that are not in the training sample.

A large-scale simulation reveals that one can solve the problem, but for languages of different language families, optimal classification algorithms and optimal features can vary greatly. The article intentionally used the simplest classifiers to comparatively test the

features used for classification. The following showed themselves to be the best (on the test sample):

- for the Russian language—random forest; intra-cluster distances from K-Means clustering; F1—0.98;
- for English—support vector machine; a combination of all signs; F1—0.98;
- for the German language—random forest; entropy-complexity; F1—0.98;
- for French—support vector machine; intra-cluster distances and averaged coordinates of cluster centres extracted from Wishart clustering; F1—0.96;
- for the Vietnamese language—random forest; entropy-complexity; F1—0.97.

Despite the absence of test-set bots in the training set and the use of simplest classifiers, a reasonable choice of features for classification made it possible to achieve a classification quality of over 96% for languages of various language families. A simulation shows that a mechanical combination of features does not necessarily improve classification quality.

Source code is available in the GitHub repository: https://github.com/quynhu-d/stb-inverse-problems (DOI: https://doi.org/10.5281/zenodo.10706994).

# FUTURE WORKS

The results of this article can be applied in various areas such as education (basic and higher), parental control, social media analysis as well as cognitive philosophy. As part of future work, we intend to further investigate our methods on different languages of different language groups and use classification models that are more effective. We also plan to use a wider range of bot models, in order to find the link between a model architecture and the sets of features, which work best for this architecture.

# ACKNOWLEDGEMENTS

The authors are indebted to Mr. J. Cumberland for text-editing and proof-reading.

# APPENDIX A. TEXT GENERATION

LSTM models were trained on human-written corpora with the following hyperparameters: batch size—16, sequence length—256, 10,000 epochs.

The remaining models were used as is from open-sources. We used different GPT-2 models for different languages from the Hugging Face. The particular models and the number of trainable parameters are in Table S1. The GPT-2 model for the Russian language is significantly larger than the models for other languages. The pretrained GPT-2 model with 124M and 356M parameters had repetition problems, so it was impossible to generate long texts. We have opted for a bigger model in order to get homogeneous texts across all languages. mGPT has 1.4B parameters, and YaLM has 1B parameters.

In order to generate the texts of similar length distribution and content as the human-written ones we employ the following procedure:

Table S2 provides details about the generated texts. The distribution of generated text lengths is similar to that of the human-written texts for the respective languages in log-scale.

**Algorithm 1:** Algorithm for text generation

**Input:**

$D$ –corpora of human-written texts in the fixed language,

*bot* –model for text generation,

$l$ –maximum number of words to generate for on prompt,

$n$ –number of texts to generate

**Output:** $D_{bot}$ –corpora of bot-generated texts

$\text{D}_{bot} \leftarrow \varnothing$

**for** $m \leftarrow 1 \ldots n$ **do**

    # Using random text from the corpora

    $d \leftarrow random\_choice(D)$

    $i \leftarrow 1$

    $d_m \leftarrow \varnothing$

    **while** $i < \ell(\text{d})$ **do**

    # *Generate* $\leq l$ *words*

    $r \leftarrow bot(d[i],l)$

    # Add them to the generated text

    $d_m \leftarrow d_m \cup r$

    $i \leftarrow i + \ell(r)$

    # Add the generated test to the corpora

    $\text{D}_{bot} \leftarrow \text{D}_{bot} \cup d_m$

**return** $\text{D}_{bot}$

## APPENDIX B. WORD EMBEDDING TECHNIQUES

The SVD method employs the singular values decomposition of the TF-IDF matrix $W$ (*Bellegarda, 2022*), constructed for a document set $D$ and a set of words $T$, with elements defined as follows:

$$w_{ij} = TF(t_i, d_j) \times IDF(t_i, D);$$

$$TF(t,d) = \frac{n_t^d}{\sum_{k \in d} n_k^d}, IDF(t,D) = \frac{|D|}{|d \in D : t \in d|},$$

$n_t^d$ denotes the number of times the word $t$ occurs in document $d$.

An $m$-rank singular decomposition of the matrix $W$ implies that one decomposes the matrix into a product of three matrices $W \simeq W' = U \Lambda V^T$. $U$ is an $M \times m$ orthonormal matrix; $\Lambda$ is a diagonal matrix, with the list of the singular values of the matrix $W$, $\lambda_1 \geq \lambda_2 \geq \cdots \geq \lambda_m > 0$, as its diagonal; and $V$ is the $N \times m$ orthonormal matrix. A hyperparameter $m$ determines the share of information one retains in the process of the singular decomposition (if $m$ is equal $min(M,N)$, one retains all information). The matrix $W'$ best approximates $W$ in $L_2$-norm, and preserves the semantic relations reflected in $W$. Row vectors of the matrix $U$ constitute word embeddings ($t_i \rightarrow u_i \Lambda$, where $u_i$ is the $i$-th row of the matrix $U$); row vectors of the matrix $V$ constitute document embeddings ($d_j \rightarrow v_j \Lambda$, where $v_j$ is the $j$-th row of the matrix $V$) (*Bellegarda, 2022*).

Advantageously, in this method, to get embeddings of various dimensions one should decompose the matrix just once. Namely, if one has obtained $m_1$-dimensional embeddings, in order to obtain $m_2$-dimensional ones, $m_2 < m_1$, one should just truncate the former to

$m_2$ components (*Bellegarda, 2022*). This significantly reduces the computational resources required to solve the problem considered: it is enough to construct embeddings for a sufficiently large value of $m$ (Most other methods for obtaining embeddings require constructing them separately for each dimension.).

In the Word2Vec method, in order to construct embeddings, one trains a specific neural network data. We train two Word2Vec architectures (we use a Python library gensim): Skip-Gram (the model, given a word, predicts its context) and Continuous Bag of Words (CBOW) (the model, given a context, predicts the word). Interestingly, Word2Vec embeddings of similar words are located close to each other in the vector space.

## APPENDIX C. ENTROPY AND COMPLEXITY

The method utilises the concept of ordinal patterns. For an $n$-gram $(x_1, \ldots, x_n)$, an ordinal pattern is defined as a permutation $\pi = (r_0, r_1, \ldots, r_{n-1})$ such that $x_{r_0} \leq x_{r_1} \leq \ldots \leq x_{r_n}$ holds. We extend the definition of ordinal patterns to multivariate time series (a sequence of n-grams) $\{x_t\}_{t=1}^n, x_t \in R^d$. For the $j$-th component ($j = 1..d$) we form the permutation $\pi_j$ as in the one-dimensional case. The general permutation for a multidimensional n-gram sequence is defined as a set of permutations $\Pi = (\pi_1, \pi_2, \ldots, \pi_d)$ (The total number of such permutations amounts to $(n!)^d$).

Next, one estimates the probability $P_i$ for $i$-th ordinal structure to occur, as the frequency of its occurrence in the multidimensional time series under consideration (a text) (1); (2) calculates two characteristics: complexity and entropy; (3) and locates the resultant point on the entropy-complexity plane in order to determine the type of series at hand. The method employs Shannon entropy:

$$S[P] = -\sum_{i=1}^{n!} P_i \cdot ln P_i, H[P] = \frac{S[P]}{S_{max}}$$

$S_{max} = S[P_e] = lnN$; $P_e = 1/N, \ldots, 1/N$ is the uniform distribution;
and MPR-complexity:

$$C[P] = Q_J[P, P_e] \cdot H[P]$$

$Q_J$ is Jenson–Shannon divergence between $P$ and $P_e$, $Q_0$ is the normalising coefficient ($0 \leq Q_J \leq 1$):

$$Q_J[P, P_e] = Q_0 \cdot (S[(P + P_e)/2] - S[P/2] - S[P_e/2]).$$

Thereby one maps a semantic trajectory (a text) into a point on the 'entropy-complexity' plane. The location of the point, with respect to the theoretical boundaries, determines the type of series (*Rosso et al., 2007*). 'Simple' stochastic processes correspond to points in the lower right corner; 'simple' deterministic ones, to points in the lower left corner; chaotic processes, to points close to the apex of the upper theoretical limit (*Rosso et al., 2007*). Figure S1 (from *Gromov & Dang (2023b)*) shows the upper and lower theoretical boundaries (for the case of a one-dimensional series); points corresponding to typical chaotic, simple deterministic, and random processes; and conditional boundaries of the respective regions.

## APPENDIX D. ATTRACTOR DIMENSIONS

The Renyi entropy of a dynamic system:

$$H_q = \frac{1}{1-q} log \left( \sum_{i=1}^{N} p(x_i)^q \right).$$

A parameter $q$ is the order of entropy; $p(x_i)$ is the probability for the state $x_i$ of the dynamical system to occur. One can interpret the Renyi entropy $H_q$ as a measure of the diversity and uncertainty of the system, with the value of $q$ influencing the sensitivity of this measure to various aspects of the probability distribution. For $q = 1$, the Renyi entropy amounts to Shannon entropy, and as $q$ changes, one gains insight on higher orders of structure and dependencies in the system.

The Renyi entropy $H_q$ is used to estimate the generalised attractor dimension of the system $D_q$. In this case, one calculates the Renyi entropy for the range $\varepsilon$:

$$\varepsilon = [ln(dist_{min}) + 1; ln(dist_{max}) - 1]$$

$dist_{min}$ is the minimum distance between two points and $dist_{max}$ is the average maximum distance between points along each data dimension. For a given $\varepsilon$, the probability for dynamical system states to be in a particular interval is:

$$H_q = \frac{1}{1-q} log \left( \sum_{i=1}^{N} p(bin(\varepsilon)_i)^q \right)$$

where $bin(\varepsilon)_i$ is the probability of the $i$-th interval of the distribution histogram. For each $\varepsilon_i$ from the interval, one calculates the entropy. Then one plots the calculated entropies $H_q$ against $\varepsilon_i$ and selects the largest linear segment of this graph in order to estimate its slope, using the least squares method. The cosine of the slope angle yields the generalised $q$-order dimension of the strange attractor $D_q$.

## APPENDIX E. CLUSTERING ALGORITHMS

Wishart clustering is a density- and graph-based algorithm. Each point is either included in a cluster or marked with noise, based on the significance of nearest clusters.

The K-Means algorithm splits objects into K clusters, minimising intra-cluster distances to cluster centroids.

C-Means is a fuzzy variation of the K-means algorithm and minimises the distances between objects and cluster centroids.

**Algorithm 2:** Wishart clustering

**Input**: $\{x_1 \dots x_l\}$—objects, $d(x_i, x_j)$—distance function, $k$, $h$

**Output**: $\left\{y_i = y(x_i)\right\}_{i=1}^{l}$—cluster labels

$d_k(x_i) \leftarrow$ distance to $k$-th nearest neighbour of $x_i$

sort objects so that $d_k(x_{(1)}) \leqslant \dots \leqslant d_k(x_{(l)})$) the cluster c is defined to be a height-significant one, with respect to height value $h > 0$ if $\max_{x_i x_j \in c}(p(xi) - p(xj)) \geq h.p(x) = \frac{k}{V_k(x)l}$, $V_k(x)$—volume of the minimum hypersphere with its centre at point $x$ containing at least $k$ observations

**for** $i \leftarrow 1$ *to* $l$ **do**
    $Vi \leftarrow x \in x(1) \dots x(i-1)|d(x(i), x) \leqslant dk(x(i))$
    $Ci \leftarrow y(x)|x \in Vi$
    **if** $Ci = \varnothing$ **then**
        generate new cluster c
        $y(i) \leftarrow$ c
    **if** $|C_i| = 1$ **then**
        let $C_i = \{c\}$
        **if** *completed(c)* **then**
            $y_{(i)} \leftarrow$ noise
        **else**
            $y_{(i)} \leftarrow c$
    **if** $|C_i| > 1$ **then**
        let $\{C_i = c_1, \dots, c_t\}$
        **if** *completed*$(c_j)$ $\forall j$ **then**
            $y_{(i)} \leftarrow$ noise
    $S_i \leftarrow c \in Ci|cissignificant(h)$
    **if** $|S_i| > 1$ **then**
        $completed(c) \leftarrow$ True $\forall c \in S_i$
        $y(x) \leftarrow$ noise $\forall x \in c \in C_i \setminus S_i$
        $y_{(i)} \leftarrow$ noise
    **else**
        merge all clusters from $C_i$ and update labels in merged clusters
**return** $y$

**Algorithm 3:** K-Means clustering

**Input**: $x_1 \dots x_l$—objects, $K$—number of clusters

**Output**: $\{ C_i, i = 1 \dots K\}$—cluster sets

generate at random $K$ centroids $c_1, \dots, c_K$

**while** *not converged* **do**
    $C_i = x : \| x - c_i \|^2 \leq \min_{j=1..K} \| x - c_j \|^2$
    $c_i = \frac{1}{|c_i|} \sum_{x \in C_i} x$
**return** $\{C_i\}$

**Algorithm 4:** Fuzzy C-Means clustering

**Input:** $x_1 \ldots x_l$—objects, $K$—number of clusters , $m$—the degree of cluster fuzziness

**Output:** $\{w_{ki} = w_k(x_i)\}_{i=1, k=1}^{l, K}$—coefficient sets with the degree of $x_i$ belonging to the $k$-th cluster

Each object $x$ is randomly assigned a cluster membership coefficient $w_k(x), \mathrm{k} = 1 \ldots K$

**while** $\| w_{(t+1)} - w_{(t)} \|^2 \geq \varepsilon (or \| C_{(t+1)} - C_{(t)} \|^2 \geq \varepsilon)$ **do**

$$c_k = [\textstyle\sum_x w_k^m(x)x]/[\textstyle\sum_x w_k^m(x)]$$

$$w_k(x) = \left[\textstyle\sum_{c \in C}(\| x - c_k \| x - c \|)^{\frac{2}{m-1}}\right]^{-1}$$

**return** $\{w_{ki}\}$

# APPENDIX F. FUZZY NUMBERS

For fuzzy clustering algorithms, a fuzzy LR number is employed (*Novak, Perfilieva & Mockor, 2012*): for a $d$-dimensional vector $x = (x_1, \ldots, x_d)$ we use a symmetric trapezoidal membership function $\mu$ (Fig. S2):

$$\mu_j(x_j) = L(\frac{m_{1j} - x_j}{l_j})[x_j \leq m_{1j}] + [m_{1j} \leq x_j \leq m_2] + R(\frac{x_j - m_{2j}}{r_j})[x_j \geq m_{2j}]$$

$m_1$ and $m_2$ are the left and right centres, respectively; $L$, $R$ are the functions of the left and right slopes; and $l$, $r$ are the width of the slopes. We assume that the $\mu_j(x_j)$ values fall on the left slope of the graph of the symmetric ($l_j = r_j$) trapezoidal function.

For each word of a given text, one constructs fuzzy numbers component by component. For the $j$-the component of a $d$-dimensional embedding $x$, we define the value of the membership function as $\mu_j(x_j) = \frac{n_j}{max_j n_j}$, where $n_j$ indicates how often one can find the $j$-th component of the vector $x$ in the text. For definiteness, we assume that $\mu_j(x_j)$ falls on the left slope of the membership function. Then, for given parameters $l_j$, $r_j$, $\Delta c = m_{2j} - m_{1j}$ (the distance between the centres $m_{1j}$, $m_{2j}$), one is able to completely restore the form of the membership function and thereby construct a fuzzy number. With the fuzzy numbers for the words thus defined, one can define the fuzzy number for an $n$-gram as the fuzzy intersection of its components, namely the minimum of the corresponding membership functions (*Novak, Perfilieva & Mockor, 2012*): $\mu((x, y)) = \{min(\mu_j(x), \mu_j(y))\}_{j=1}^d$.

The Euclidean distance between two fuzzy numbers is defined as:

$$d(\widetilde{x}_i, \widetilde{x}_j) = ||m_{1i} - m_{1j}||^2 + ||m_{2i} - m_{2j}||^2 +$$

$$||(m_{1i} - \lambda l_i) - (m_{1j} - \lambda l_j)||^2 + ||(m_{2i} + \rho r_i) - (m_{2j} + \rho r_j)||^2)^{1/2}$$

# APPENDIX G. DATA DISTRIBUTION

Length distributions of texts (in words) are provided in Figs. S3–S5.

## APPENDIX H. ENTROPY-COMPLEXITY FOR OTHER LANGUAGES

Points corresponding to texts of English, German, French and Vietnamese texts on the entropy-complexity plane are provided in Figs. S6–S9.

## APPENDIX I. ACCURACY SCORE VALUES

Accuracy score values for classification with entropy-complexity measures, semantic trajectory characteristics, Wishart clustering-based classifiers, K-Means clustering-based classifiers, classifiers based on intra-cluster distances and unified classification model are provided in Tables S3-S8.

## APPENDIX J. GENERALISED ENTROPY VALUES

Generalised entropy values for texts of the English and Russian languages are provided in Figs. S10–S11.

### Funding

The publication was prepared within the framework of the Academic Fund Program at HSE University (grant No 24-00-040 Spot the Bot: the Inverse Problems of NLP). The study made use of the computational resources of HPC facilities at HSE University. The funders had no role in study design, data collection and analysis, decision to publish, or preparation of the manuscript.

### Grant Disclosures

The following grant information was disclosed by the authors:
Academic Fund Program at HSE University: 24-00-040.

### Competing Interests

The authors declare there are no competing interests.

### Author Contributions

- Vasilii A. Gromov conceived and designed the experiments, authored or reviewed drafts of the article, and approved the final draft.
- Quynh Nhu Dang conceived and designed the experiments, performed the experiments, analyzed the data, performed the computation work, prepared figures and/or tables, authored or reviewed drafts of the article, and approved the final draft.
- Alexandra S. Kogan conceived and designed the experiments, performed the experiments, analyzed the data, performed the computation work, prepared figures and/or tables, authored or reviewed drafts of the article, and approved the final draft.
- Assel Yerbolova performed the experiments, performed the computation work, authored or reviewed drafts of the article, and approved the final draft.

## Data Availability

The source code is available at GitHub and Zenodo:

- https://github.com/quynhu-d/stb-inverse-problems.

- nina. (2024). quynhu-d/stb-inverse-problems: v1.0.0 (v1.0.0). Zenodo. https://doi.org/10.5281/zenodo.10706994.

## Supplemental Information

Supplemental information for this article can be found online at http://dx.doi.org/10.7717/peerj-cs.2550#supplemental-information.

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
