# Peer review of "Spot the bot: the inverse problems of NLP"

_PeerJ Computer Science, doi:10.7717/peerj-cs.2550_

## Round 0.1 · original submission · Major Revisions

Dear authors,

Thank you for your submission. Feedback from the reviewers is now available. Your article has not been recommended for publication in its current form. However, we do encourage you to address the concerns and criticisms of the reviewers and resubmit your article once you have updated it accordingly.

Best wishes,

Reviewer 1 ·

Basic reporting

1. The author could consider adding and discussing the current literature on graph-based methods applied to bot detection, along with their advantages and potential limitations.
2. The author should ensure that the code link provided on GitHub is valid so that other researchers can access and reproduce the experimental results. Should the link have changed, the documentation should be updated to reflect the new URL.

Experimental design

3. To improve the paper's quality and trustworthiness, it is essential that the author provides a more comprehensive description of the models and methodologies employed for generating bot-generated texts. Details should encompass the rationale behind the selection of specific bot generation models, the intricacies of the training process, the parameter configurations for text generation, and other pivotal particulars. The caliber of bot-generated texts holds paramount importance for the relevance and impact of bot-related research. Elucidating these aspects not only facilitates a deeper understanding for readers but also ensures the reproducibility of the research and propels further scholarly inquiries.

Validity of the findings

No comment.

Cite this review as
Anonymous Reviewer (2024) Peer Review #1 of "Spot the bot: the inverse problems of NLP (v0.1)". PeerJ Computer Science

Reviewer 2 ·

Basic reporting

- There are many grammatical errors in the paper, they should be corrected.
- The meaning of abbreviations used for the first time should be given.
-There are errors in the texts in Figure 1. They should be redrawn.
- No preliminary information about the problem is given in the summary section of the paper. There is no demonstration of the presented method. A summary of the studies carried out throughout the article should be given in this section.
- A flowchart of the method should be added to the paper.
- Characteristics of texts written by humans are presented in Table 1. Details about the produced texts should also be given in a table.

Experimental design

- Why were the performances of the algorithms used in the proposed method evaluated only in terms of F1-score, why were other performance metrics not examined, for example, accuracy values can be added to the experimental results?
- Why are the results for the Random Forest algorithm not given for the experimental results with Semantic trajectory characteristics, Wishart clustering-based classifiers and K-Means clustering-based classifiers?
- The conclusions section is written in bullet points. It should be discussed whether studies on this subject will continue and if so, what contributions or innovations can be added (Future works). Researchers should be guided by indicating in which areas the proposed method can be used.

Validity of the findings

- As stated by the authors themselves, this study is a continuation of two previous studies they conducted on the same subject. When the two articles mentioned are examined, it is seen that they are quite similar to this study. The experiments that could be given within the scope of one study were intended to be divided into 3 articles. Reducing the similarity rate by changing the words does not mean that the articles are not similar. Even some figures in the methods section are taken directly from the previous article (Figures 1 and 2).
- What is the contribution and uniqueness of the study to the literature? It should be noted that it is different and superior to previous studies. As in the abstract, no information about the method is included in the introduction.

Additional comments

In this article, a study has been carried out on a subject that has attracted a lot of attention, especially in recent years. However, there are major shortcomings in the article. The article must be examined according to the reviewer's evaluations.

Cite this review as
Anonymous Reviewer (2024) Peer Review #2 of "Spot the bot: the inverse problems of NLP (v0.1)". PeerJ Computer Science

---

## Round 0.2 · accepted · Accept

Dear Authors,

Thank you for the revised paper. One of the reviewers who had been invited to comment on the previous version of the manuscript did not respond. However, the other reviewer accepted the paper as it stood. I am also satisfied with the current version and believe that this manuscript is ready for publication.

Best wishes,

Reviewer 2 ·

Basic reporting

The authors have made all suggested revisions. The quality and clarity of the paper have improved, and it is acceptable as it stands.

Experimental design

-

Validity of the findings

-

Additional comments

-

Cite this review as
Anonymous Reviewer (2024) Peer Review #2 of "Spot the bot: the inverse problems of NLP (v0.2)". PeerJ Computer Science